# An exploratory study of behavioral traits and the establishment of social relationships in female laboratory rats

Shiomi Hakataya[1,2,3☯], Noriko Katsu[1,4☯], Kazuo Okanoya[1,2]*, Genta Toya[1,2]*

1 Graduate School of Arts and Sciences, The University of Tokyo, Meguro, Tokyo, Japan, 2 Advanced Comprehensive Research Organization, Teikyo University, Itabashi, Tokyo, Japan, 3 Japan Society for the Promotion of Science, Chiyoda, Tokyo, Japan, 4 Graduate School of Human Sciences, Osaka University, Suita, Osaka, Japan

☯ These authors contributed equally to this work.
* toyagent@protonmail.com (GT); kazuookanoya@gmail.com (KO)

## Abstract

There is growing evidence that social relationships influence individual fitness through various effects. Clarifying individual differences in social interaction patterns and determinants for such differences will lead to better understanding of sociality and its fitness consequences for animals. Behavioral traits are considered one of the determining factors of social interaction. The purpose of this study was to explore the effects of individual behavioral traits on social relationship building in laboratory rats (*Rattus norvegicus*), a highly social species. Initially, the following behavioral characteristics were measured in individuals: tameness (glove test), activity (open field test), exploration (novel object test), sociability (three-chamber test), and boldness (elevated plus maze test). We then used DeepLabCut to behaviorally track three groups of four individuals (12 total) and analyze social behaviors such as approach and avoidance behaviors. Principal component analysis based on behavioral test results detected behavioral traits interpreted as related to exploration, boldness, activity, and tameness, but not sociability. In addition, behavioral tracking results showed consistent individual differences in social behavior indices such as isolation time and partner preference. Furthermore, we found that different components were correlated with different phases of social behavior; exploration and boldness were associated with the early stages of group formation, whereas activity was associated with later stages of relationship building. From these results, we derived hypothesize that personality traits related to the physical and social environment have a larger influence in the relationship formation phase, and the behavioral trait of activity becomes important in the maintenance phase of relationships. Future studies should examine this hypothesis by testing larger group sizes and ensuring there is less bias introduced into group composition.

## Introduction

The formation of social relationships with conspecifics is one of the most important factors affecting the fitness of gregarious animals. Social relationships can be viewed as the result of a

**Data Availability Statement:** The relevant data and code are available at the following link: https://osf.io/nk6h7/?view_only=d5b3df2fe2004be289add98180faa40a.

**Funding:** This work was supported by Japan Society for the Promotion of Science (JSPS: https://www.jsps.go.jp/english/index.html) Grant-in-Aid for Scientific Research on Innovative Areas Grant Number JP17H06380 (KO), JSPS Grant-in-Aid for Scientific Research (A) Grant Number JP20H00105 (KO), JSPS Grant-in-Aid for Scientific Research (S) Grant Number JP23H05428 (KO), JSPS Grant-in-Aid for Transformative Research Areas Grant Number JP21H05175 (GT) and JSPS Grant-in-Aid for JSPS Fellows Grant Number JP21J23113 (SH), JP22KJ0702 (SH), JP19K20643 (NK), JP21J40080 (NK), JP20J01616 (GT). The funders had no role in study design, data collection and analysis, decision to publish, or preparation of the manuscript.

**Competing interests:** The authors have declared that no competing interests exist.

series of social interactions among individuals and are characterized by context (e.g., sexual, parenting, affiliation, or conflict), quality (e.g., intensity of affiliation or aggression), and temporal patterns (e.g., frequency) [1]. Strong affiliative social relationships between same-sex individuals can bring a variety of short-term benefits (e.g., grooming, reduced competition) as well as long-term benefits (e.g., survival) to the focal individuals [2–4]. For instance, social bonds have been reported to have positive impacts on longevity in female baboons (*Papio hamadryas ursinus*) [5] and reproductive success in female feral horses (*Equus caballus*) [6], female house mice (*Mus domesticus*) [7] and female baboons [8]. Thus, to better understand the social life of animals and its fitness consequences, it is essential to clarify how individuals choose social partners and form social relationships with them. Factors such as kinship (ring-tailed coatis, *Nausa nasua* [9]; primates [10]), sex (cercopithecines [11]), similarity in age (mouflon, *Ovis gmelina* [12]) and in social rank (rhesus monkeys, *Macaca mulatta* [13]) are known to affect social preferences. In addition, there is accumulating evidence that personality traits affect social relationships in nonhuman animals.

Personality (or temperament) refers to individual differences in behavior that are consistent across time and context [14]. Many approaches have been developed to characterize personality in non-human animals. For example, in a systematic bottom-up approach, the animal species' universal behavior repertoire is comprehensively analyzed to extract personality dimensions [15]. On the other hand, many studies adopt top-down approaches, aiming to describe the personality of the target species based on personality dimensions already reported in other species or in other studies. In primates, most commonly identified personality dimensions include dominance, excitability, sociability, and activity, which are measured through behavioral coding and observer trait rating by questionnaires [16]. In non-primate animal research, five main personality axes have been proposed and extensively studied: boldness, exploration, activity, aggressiveness, and sociability [17]. Many of these personality traits are known to influence the social interaction patterns of the focal animals [18]. Given that the major personality dimensions may differ among animal species [15], the approach based on these axes may be too simplistic and other important traits may be missed. Still, these axes can be viewed as a valuable basic framework which provides a wealth of comparability with results obtained in other studies. Thus, our study employed boldness, exploration, activity and sociability among these axes.

Boldness represents an individual's reaction to risky situations and can be measured by the presentation of threating stimuli such as predators [17]. Studies in various species have shown that bold individuals tend to be less sociable than shy ones (eastern grey kangaroos, *Macropus giganteus* [19]; sheep, *Ovis aries* [20]; great tits, *Parus major* [21]; Eastern garter snakes, *Thamnophis sirtalis sirtalis* [22]). On the other hand, some reports show a positive correlation between sociability and boldness (Deccan mahseer, *Tor khudree* [23]), while others show no correlation (female guppies, *Poecilia reticulata* [24]; three-spined sticklebacks, Gasterosteus aculeatus [25]).

Exploration describes an individual's reaction to novel situations and can be evaluated through tests presenting a novel environment, object, or food [17]. Activity corresponds to the general level of activity of an individual expressed in terms of distance traveled in the open field test [17]. As for these traits, exploration was positively correlated with sociability in birds [26], while activity was positively correlated with sociability in yellow-bellied marmots (*Marmota flaviventris*) [27]. Moreover, positive correlations between all three traits (sociability, activity and exploration) were seen in the delicate skink (*Lampropholis delicata*) [28]. However, negative correlations (rabbits, *Oryctolagus cuniculus* [29]) and no correlations (male starlings, *Sturnus vulgaris* [30]) between exploration and sociability have also been demonstrated.

Sociability represents an individual's tendency (frequency/intensity) to interact with conspecifics, excluding aggressive or agonistic behaviors [17]. Sociability can be assessed naturally by observing the social relationships that an individual has within a free-ranging group or by behavioral tests that measure responses to social stimuli presented in standardized environments [18]. Sociability, as assessed experimentally by behavioral tests, has been found to be related to social behavior observed naturally; more sociable individuals in the personality assay are actually more attracted to conspecifics in free interactions (Eastern garter snakes [22]; three-spined sticklebacks, [31]).

Similarity of personality also modulates social relationships. Individuals with similar sociability have strong social bonds in chimpanzees (*Pan troglodytes* [32]), and highly affiliative relationships in brown capuchin monkeys (*Sapajus* sp. [33]) and bonobos (*Pan paniscus* [34]).

As described above, the influence of personality traits on social relationships (i.e., which traits affect social relationships and how) varies greatly between studies and species. To understand the relationship between personality traits and social interaction patterns in a variety of animal societies, further investigation is needed. Particularly, comprehensive consideration of multiple personality traits is required.

The laboratory rat is domesticated from the Norway rat (*Rattus norvegicus*) [35]. In the wild, rats form complex societies consisting of up to several hundred individuals which may contain small sub-groups [36, 37]. Rats are highly social and have demonstrated various forms of prosocial behaviors in a laboratory setting (e.g., preferentially providing food rewards to a partner [38]; freeing a trapped partner by opening the door [39]; reciprocal cooperation of allogrooming and food provisioning [40]; for review, see [41]). Also, rats are known to possess high social recognition abilities such as individual recognition [42] and kin recognition [43].

Given their social characteristics, social relationships in rat groups can be complex and there may exist social partner preferences. Whether rats form relationships with conspecifics based on partner preference remains unclear. Proops et al. [44] found that male fancy rats (rats of companion animals) associated non-randomly and most individuals had significantly more preferred/avoided partners compared to chance. However, Schweinfurth et al. [45] reported that no stable social bonds were observed in groups of female wild-derived rats. To clarify whether rats have social bonds, there are several challenges to be solved. First, the resolution of social relationship analysis needs to be improved. Particularly, sampling rates of behavioral measures should be as high as possible. Second, previous studies mainly focused on proximity and bodily contact between individuals at a specific moment. We should be able to deepen our understanding of social relationships by considering the direction of social behavior that alters inter-individual distance (i.e., approaching vs. departing), not just by measuring the consequence of these behaviors (i.e., proximity vs. isolation). Finally, the process of social bond formation is not described, since previous studies examined social relationships within established groups. Thus, we need to examine how social relationships (and social bonds) are first established by joining previously unfamiliar individuals together.

Personality traits such as boldness have been shown to affect social relationships in a variety of species. Although laboratory rats are considered model animals, they do show stable inter-individual differences in behavior [46]. However, behavioral traits in rats and how these traits are associated with social behaviors remain understudied. Therefore, it is necessary to examine behavioral traits more comprehensively, including traits in the social domain and how these traits are related to social relationships. As in previous studies, characteristics such as boldness, exploration, and sociability are associated with social relationships but may have different effects during the formative stages of social relationships vs. after they are fully established. Furthermore, since laboratory rats are domesticated animals with increased tameness, tameness may also affect social behavior toward other individuals.

Among domestic rodent studies, it has been shown that domestic guinea pigs (*Cavia porcellus*) are less aggressive and more tolerant of conspecifics than their wild counterparts [47, 48]. Also, studies with rats and mice have shown that selective breeding for tameness can be achieved using handling tests, suggesting that individual differences in tameness can be measured even within the same strain maintained in the laboratory [49, 50]. If individual differences in tameness exist in laboratory rats, this could affect how much the individual tolerates proximity/approach by other conspecifics.

In this study, we aimed to exploratorily investigate the individual differences in social relationships of laboratory rats, and the effects of behavioral traits on these individual differences. First, we evaluated behavioral traits of laboratory rats by performing a series of standardized behavioral tests. Considering the major personality axes in non-primate animal research [17], we chose behavioral tests which measured boldness (elevated plus maze test), exploration-avoidance (novel object test), activity (open field test), and sociability (three-chamber test). In addition, we measured tameness using the glove test. Caution should be exercised when interpreting the traits measured in these tests. For instance, there can be some overlap between the tests in traits they measure, and the same test is used to measure different traits depending on the study [51]. Thus, it is quite unlikely that there are one-to-one correspondences between behavioral tests and personality traits, so we performed principal component analysis to interpret the traits measured by the tests. Later, we conducted behavioral tracking of freely behaving rats using DeepLabCut [52] to examine whether rats showed individual differences in social relationships and how differences in personality traits affected the animals' social behavior toward familiar vs. unfamiliar conspecifics. Behavioral tracking was conducted over two time periods: during the first encounter, and once the relationships were considered stable. For each time period, we examined the consistency of the partner choice during social interactions. We then investigated the correlation between measured personality traits and social interaction, particularly with familiar vs. unfamiliar conspecifics. We also examined whether the degree of personality similarity affected partner preference.

## Materials and methods

### Approval for animal experiments

All procedures were conducted in accordance with the experimental implementation regulations of the University of Tokyo. This study was approved by the animal experimental committee at the University of Tokyo, Graduate School of Arts and Sciences (Permission Number: 2021–2).

### Subjects

Subjects were female Sprague-Dawley (SD) rats, descended from three pregnant females (Mothers A-C) purchased from a vendor (SLC Japan Inc., Hamamatsu, Japan). Rats were born and raised in our laboratory. Throughout the whole experiment, animals were kept under a 12:12h light/dark cycle (lights on at 8:00 am) and had *ad libitum* access to food (Lab Diet, PMI Nutrition International, St. Louis, U.S.A.) and water. Eighteen rats were used for the behavioral tests (five individuals from Mother A, seven from Mother B, and six from Mother C, respectively) and twelve rats were used for behavioral tracking (three individuals from Mother A, five from Mother B, and four from Mother C, respectively). We used solely female rats because we aimed to examine social interactions between same-sex individuals outside of breeding context.

## Experimental design

The timeline of the experiment, housing conditions, and animals' ages are depicted in Fig 1. After weaning at three weeks of age, animals were housed in pairs (pair housing 1). Rats were assigned to pairs in such a way that there were equal numbers of sibling and non-sibling pairs.

First, we assessed behavioral traits of the individuals using a series of behavioral tests when rats were approximately two months of age (Fig 1). Behavioral tests included the glove test, open field test, novel object test, three-chamber test and elevated plus maze test. Eighteen rats were used. From approximately three months of age, sixteen out of eighteen rats were housed in groups of four individuals (group housing 1) (Fig 1). The aim of this was to acclimate the rats to group housing (both the tracking cage itself, and the group dynamics) prior to behavioral tracking (group housing 2). Two individuals were not used to match population size. To form the groups, existing pairs were combined in such a way that each group of four had at least one individual from each mother. Animals were individually marked with dots on the back and stripes on the tail by a blue animal marker filled with food dye (FG 2200 series; Muromachi Kikai, Tokyo, Japan).

Second, we investigated social interaction patterns among group members of four individuals through behavioral tracking for one week (term 1 of group housing 2) (Fig 1). Two pairs were randomly chosen and removed from the subjects at this stage, to use for another experiment. The remaining twelve rats were used for behavioral tracking. The groups were named Hippo, Okapi and Panda. Rats were approximately seven months of age at this point. To examine how rats familiarize with new cage-mates, group members were exchanged

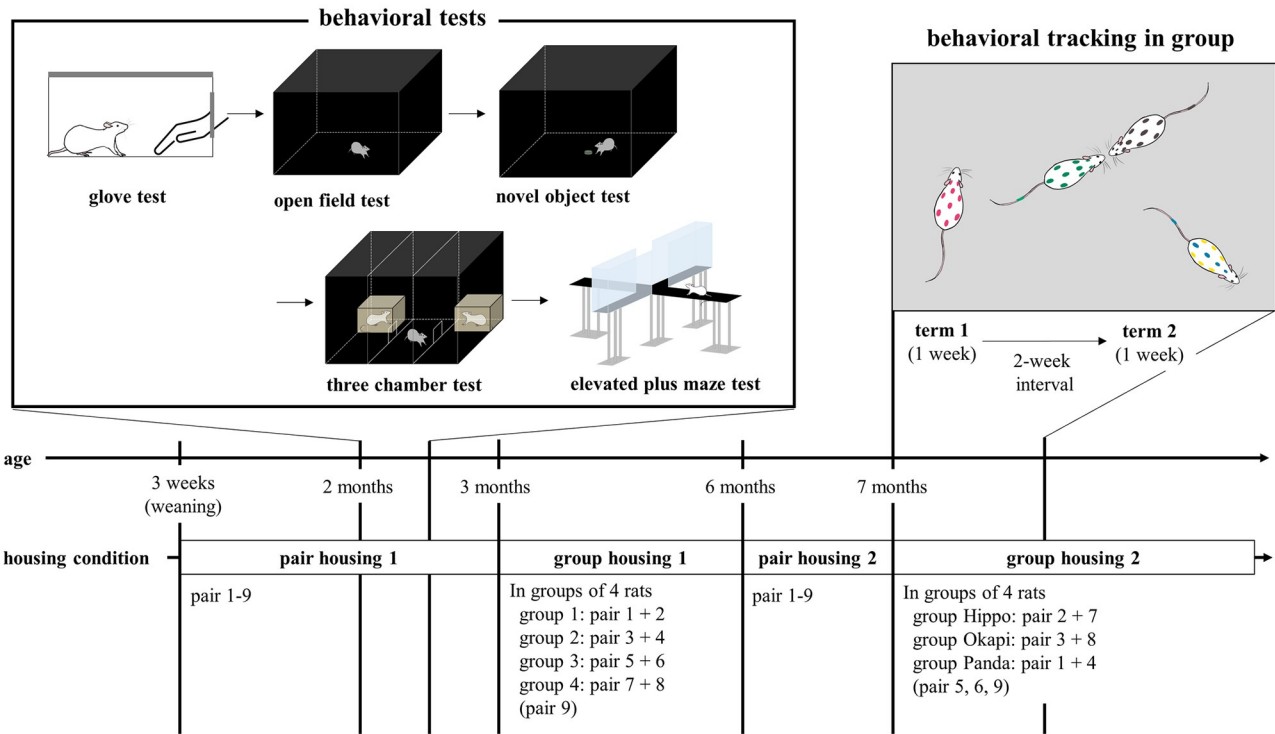

**Fig 1. Experimental design.** During pair housing 1, we assessed the behavioral traits of individuals using a series of behavioral tests when rats were approximately two months of age. Afterwards, rats were housed in groups of four individuals to acclimate them to group housing (group housing 1). At approximately seven months of age, we exchanged group members and investigated social interaction patterns among individuals through behavioral tracking for one week (term 1 of group housing 2). After two weeks, behavioral tracking was repeated (term 2 of group housing 2).

between acclimation groups (group housing 1) so that each group included both familiar and unfamiliar individuals at the start of behavioral tracking. After a two-week rest interval, behavioral tracking was repeated for one week with the same groups to test the stability of social interaction patterns (term 2 of group housing 2). To aid in identification for tracking, the animals were marked with a polka-dot pattern from the forehead to the base of the tail. Each rat was marked with a different color combination (blue and either yellow, red, green, or black).

## Behavioral tests

We conducted the glove test, open field test, novel object test, three-chamber test and elevated plus maze test. All individuals experienced the tests in the same order as listed above. This order was chosen so that rats started with the least stressful test, and ended with the most stressful test, with the level of stress increasing gradually. The interval between tests was 1–2 days. Testing animal order was randomized across pairs. Two animals from the same pair were tested on different days except for the novel object test, so that the testing experience of one rat would not influence the other one. Before the tests, subject animals were individually transferred to the testing room in a small cage. All tests were conducted during the light phase, with no deprivation conducted before the experiment. All experimental apparatuses for behavioral tests were cleaned with 75% ethanol before each trial. Behavioral tests were recorded using a monitoring camera (BSW500MBK, Buffalo, Nagoya, Japan).

Additionally, the food allocation test was conducted to determine dominance relationships between subjects (modified after Ziporyn & McClintock [53]). We placed a high value food item (a piece of apple) on the lid of the home cage of each pair and observed the pair of rats (under at least four hours of food deprivation) attempt to displace the food. The rat that monopolized the food was deemed the winner. However, we did not find clear dominance relationships; thus, the data was not used for subsequent analysis.

In the analysis, we were not able to use behavioral indices which depended on time due to a recording error (the number of frames per second varied within the same video), except for the glove test. We therefore used behavioral indices based on distance traveled and countable behaviors in each test. Also, because recording durations varied among individuals in the open field and three-chamber tests, the distance traveled and the number of entries were divided by the total recording duration for these tests. Recorded videos of the behavioral tests were analyzed using a tracking software, ANY-maze (Stoelting, Wood Dale, U.S.A.), except for the glove test.

**Glove test.** The glove test measures tameness of a subject animal when presented with a human hand covered with double cotton gloves. We conducted the glove test following Albert, et al. [49]. The testing apparatus was a standard rodent cage (42cm length × 25cm width × 18cm height) with an opening (10cm × 10cm) on the wall. For more details, see S1 File.

Recorded videos of the glove test were scored using tanaMove_v0.09 (event recording software by keyboard typing). The total duration of approach, avoidance, and handling were measured. No aggressive behaviors were observed during the test.

**Open field test.** The open field test measures anxiety-related emotional behavior and locomotor activity of the subject animal. The testing apparatus was a black acrylic square arena (80cm length × 80cm width × 50cm height), with an LED light above the center (360 lx at the center, 280 lx around the edges) to provide even lighting. Animals were placed into the center of the arena and allowed to freely explore for five minutes.

The open field arena was divided into a 5 × 5 grid of 16cm squares. The central nine blocks were defined as the central zone. The total distance traveled and the total number of entries into the central zone were calculated. Total recording durations ranged from 299 to 324 s.

**Novel object test.** The novel object test measures neophobia/neophilia of the subject animal. The testing apparatus was the same square arena which was used for the open field test. The test consisted of three phases: habituation, interval and object exploration. At first, animals were placed into the center of the front side of the arena and released for a five-minute habituation period. Next, animals were taken out of the open field, returned to the small cage and remained there for two minutes. During this period, the arena was cleaned (if necessary) and the novel object (2g of chocolate coated barley puffs in a small ceramic bowl) was placed in the middle of the arena. Finally, animals were returned into the center of the front side of the arena facing the novel object and released for a five-minute exploration period. We used chocolate puffs, which have potentially high values for rats, because approaching behaviors were hardly observed in a preliminary experiment with non-subject rats using neutral objects.

As in the open field test, the arena was divided into a 5 × 5 grid of 16cm squares. The central nine blocks were defined as the central zone and the central one block was defined as the object zone. The total distance traveled, the total number of entries into the central zone, and the total number of times that the animal's head entered the object zone were calculated. To control for individual baseline activity, differences between the habituation phase and test phase were calculated for each value. No food consumption was observed during the test.

**Three-chamber test.** The three-chamber test measures sociability of the subject animal. The testing apparatus was a black acrylic square arena (80cm length × 80cm width × 50cm height) divided into three compartments by transparent acrylic partitions, with two restraint chambers placed in each side compartment. The partitions had a square opening (10cm × 10cm) at the center bottom for animals to pass through. Restraint chambers were wire cages with plastic bottoms (18cm length × 23cm width × 20cm height) which allowed social interaction between subject and stimulus animals. The test consisted of three phases: habituation, interval and social exploration. First, animals were placed into the front end of the center rectangular compartment and released for a five-minute habituation period. Second, animals were taken out of the arena, returned to the small cage and remained there for two minutes. During this period, the arena was cleaned (if necessary) and a social stimulus (unfamiliar conspecific) and an inanimate white rat toy (head and body length: approximately 19cm long) were placed in either end of the restraint chambers of the side compartments (counterbalanced). Finally, animals were returned into the center rectangular compartment and released for a five-minute social exploration period.

The total distance traveled and the total number of entries into the compartment with the stimulus animal and the inanimate rat toy were calculated for both the habituation period and the social exploration period. Then, differences between the habituation phase and test phase were calculated for the distance traveled per time. For the number of entries to each compartment, preference for the animal compartment was calculated for both the habituation phase and phase as follows: the total number of entries into animal compartment was divided by the total number of entries into both compartments. To control for individual left-right bias, the differences between the habituation phase and test phase were used for the analysis. Total recording duration ranged from 275 to 349 s.

**Elevated plus maze test.** The elevated plus maze test measures anxiety of the subject animal. The testing apparatus was a black acrylic maze which consisted of four arms (50cm length × 10cm width) and a center area (10cm length × 10cm width). Two arms were open without walls and the other two were enclosed by transparent acrylic walls (45cm height). The maze was elevated 58cm off the floor. Animals were placed in the center area facing an

open arm (and facing away from the experimenter) and allowed to freely explore for five minutes.

The total distance traveled, the total number of entries into the open arms, and the total number of entries into the closed arms were calculated.

### Behavioral tracking in a group

Behavioral tracking was carried out in large wire cages (91cm length × 54cm width × 154cm height). Each tracking cage contained recycled paper chip bedding, two shelters, and a feeding bowl. Cage walls were covered by polypropylene foam sheets to prevent rats from climbing up the walls. Food and water were available *ad libitum*. A camera (HERO10 BLACK CHDHX-101-FW, GoPro, San Mateo, U.S.A.) was attached to the ceiling of the cage. Each tracking cage was placed in a separate room under a 12:12h light/dark cycle (lights on at 8:00 am). During the dark phase, indirect dim lights (approximately 20 lx) were set to allow behavioral tracking using color markings. All subjects had experienced housing in the tracking cage during the initial group housing of four individuals with different cage-mates (Fig 1).

Prior to behavioral tracking, animals were habituated to tracking cages per pair for two nights. On the first day of the behavioral tracking, animals were introduced to tracking cages at least five hours before the start of behavioral recording and stayed there for seven days (seven tracking sessions). Each tracking session was nine hours in duration (21:00–6:00) under dim light. For reliable color tracking, shelters were removed at least one hour before the start of each tracking session and set again the next morning. Animals were left undisturbed from the time shelters were removed to the time they were reintroduced. To enable identification for tracking, a polka-dot pattern marking using colored markers was reapplied at least five hours before the start of each tracking session. After a two-week interval, behavioral tracking was repeated for seven days to test the stability of social interaction patterns.

We were able to obtain video data for 14 days for the Panda group, 13 days for the Hippo group (day 4 of term 2 was excluded from the data due to camera malfunction), and 12 days for the Okapi group (days 4 and 5 of term 1 were not recorded). Individual location (X and Y coordinates in the video frame) was estimated using DeepLabCut 2.2 and accuracy was improved using a self-developed correction program (see OSF for the code). Estimation accuracy test results showed that on almost all dates, the estimation accuracy was 96% or higher (see S1 Table). Based on the estimated X and Y coordinates, indices for social interaction (time in proximity, time in isolation, number of approaches and number of avoidances) were calculated. For more details, see S1 File.

### Statistics

We conducted principal-components analysis based on thirteen items from five behavioral tests by using the "prcomp" function in R version 4.1.0 [54]. Each component was interpreted by assigning items with salient loadings ($\geq 0.4$) [55]. If the same item was salient in more than one component, the component with higher loadings was considered.

We used nine indices for social interactions. Total time in isolation, and time in proximity with familiar and unfamiliar individuals were indices of sociability and partner preference, respectively. Total number of approaches, and avoidances from familiar and unfamiliar individuals were indices of the amount of social interaction. In a group of four, each subject was paired with two unfamiliar individuals and one familiar individual that the subject was raised with. Thus, the indices related to unfamiliar group members were calculated using the average value for the two unfamiliar individuals. We used the average value of seven days of each term for each index. Days 4 and 5 in term 1 for group Okapi and day 4 in term 2 for group Hippo

were excluded from the analyses as mentioned above. As we found the significant group differences in time spent in proximity to other members (see S2 Table), all indices were standardized by the average value for each group.

The consistency of individual differences in social interaction was quantified using intraclass correlation coefficients (ICC [3,k]), using the R package "psych". ICC was calculated for time in isolation of each subject per group, and for time in proximity to each member per subject. We judged substantial consistency as ICC values more than 0.7 [56]. We then examined whether the preference for a partner was explained by familiarity. Linear mixed model (LMM) and likelihood ratio tests were conducted using the R package "lme4". The response variable of the LMM was time in proximity to each member, the explanatory variable was familiarity with each member, and random variables were the subject IDs. The effect of familiarity was examined using a likelihood ratio test to compare the full model with a model that did not include familiarity as a factor.

We examined whether a measured personality trait was related to actual behavior in a naturalistic social group. We calculated the correlation coefficients between each principal component score of the subjects and the nine indices for social interaction.

Finally, we examined whether pairs that have similar behavioral traits were more likely to form affiliative relationships. The absolute difference between the principal component scores of two individuals was used as an index of personality trait similarity. For a given pair, we calculated the correlation between the similarity scores for the first to fourth principal components and time spent in proximity. Indices related to approach and avoidance behavior were not included in this analysis because they were directional indices.

## Results

### Personality structure

The principal component analysis was conducted based on 13 variables. The Kaiser criterion supported the adoption of the first three components (S1 Fig), however, we eventually adopted four components due to the novelty of the component. Four components explained 75.3% of the total variance. The first component explained 29.7% of the variance and was defined by high loadings for center area entry and total distance traveled in the novel object test (Table 1). This component was interpreted as an exploration-avoidance axis, as high-scoring subjects showed a tendency to actively explore in a novel environment.

The second component was defined by positive loadings for open area entry of the elevated plus maze test. This was interpreted as a shyness-boldness axis, as the high-scoring subjects showed a tendency to move more in anxiety-evoking situations.

The third component was characterized by the open field test. PC3 exhibited negative loadings for total distance traveled and center area entry in the open field test. Considering the content of PC2, PC3 was interpreted as activity. High-scoring individuals showed less activity level in general.

The fourth component was manifested by positive loadings for time spent handled in the glove test. This component also showed high loadings for the number of center area entries in the open field test. PC4 was interpreted as tameness, as high-scoring individuals were more likely to tolerate human handling.

The items measured in the three-chamber test, and in particular the number of entries into the animal chamber, did not have high loadings in PC1 to PC5 (Table 1). This suggests that sociability measured by this test did not explain the large variance of behavioral characteristics.

**Table 1. Personality traits measured by the five behavioral tests.**

| Item | PC1 | PC2 | PC3 | PC4 | PC5 |
|---|---|---|---|---|---|
| NO center area entry | **0.401** | 0.288 | 0.107 | -0.086 | -0.162 |
| NO distance | **0.413** | 0.265 | 0.092 | -0.107 | 0.138 |
| NO head entry | 0.376 | 0.341 | -0.037 | -0.167 | -0.111 |
| EPM open area entry | -0.279 | **0.416** | 0.093 | 0.045 | -0.323 |
| EPM close area entry | -0.389 | 0.307 | 0.186 | -0.133 | -0.108 |
| EPM distance | -0.337 | 0.382 | 0.201 | -0.128 | -0.040 |
| OF distance | -0.042 | 0.322 | **-0.561** | 0.063 | -0.011 |
| OF center area entry | 0.080 | 0.232 | **-0.492** | <u>0.418</u> | 0.265 |
| GL handling | -0.026 | 0.126 | 0.134 | **0.731** | -0.310 |
| GL avoid | 0.134 | -0.012 | -0.358 | -0.335 | **-0.577** |
| GL approach | -0.276 | 0.079 | -0.247 | -0.301 | 0.287 |
| TC distance | -0.066 | -0.369 | -0.185 | 0.026 | **-0.493** |
| TC animal area entry | 0.278 | 0.032 | 0.310 | 0.040 | -0.007 |
| Proportion of variance | 0.297 | 0.211 | 0.150 | 0.095 | 0.075 |

NO: novel object test, EPM: elevated plus maze test, OF: open field test, GL: glove test, TC: three-chamber test. The items with loadings larger than ± 0.4 are highlighted in boldface. For the items in which there were high loadings in more than one component, the lower loadings are underlined.

## Individual differences in social interaction

Consistency of the time spent in isolation tended to vary by group (Table 2). The ICCs were greater than 0.7 for both terms 1 and 2 in the Hippo group; therefore, in this group, the rank order of time spent in isolation among group members was stable. The ICCs were moderate

**Table 2. Consistency of individual differences in time spent in isolation and time spent in proximity to other cage mates.**

| Group | ID | Term1 | Term2 |
|---|---|---|---|
| *Time in isolation* | | | |
| Hippo | | **0.760** | **0.720** |
| Okapi | | 0.000 | 0.000 |
| Panda | | 0.447 | 0.589 |
| *Proximity with other cagemates* | | | |
| Hippo | A3 | **0.910** | **0.710** |
| Hippo | C4 | **0.876** | **0.799** |
| Hippo | B5 | 0.000 | 0.363 |
| Hippo | B6 | 0.240 | 0.000 |
| Okapi | A4 | 0.496 | **0.822** |
| Okapi | C5 | 0.690 | 0.690 |
| Okapi | C1 | **0.755** | **0.774** |
| Okapi | C2 | 0.000 | 0.000 |
| Panda | A1 | 0.000 | **0.780** |
| Panda | B7 | 0.000 | 0.204 |
| Panda | B1 | 0.000 | 0.484 |
| Panda | B2 | 0.000 | 0.000 |

Intraclass correlation coefficients (ICC [3,k]) for social behavior were calculated for each term, which included seven days of data. A value more than 0.7 was judged as substantially consistent and highlighted in boldface.

(0.4–0.7) for both terms 1 and 2 in the Panda group. No consistent individual differences in isolation were identified throughout the period in the Okapi group. Therefore, with the exception of one group, individual differences in sociability, as measured by a preference for proximity to others, were found in our subject female rats.

The proximity to each group member did not necessarily show the same tendency as time spent in isolation. The three out of twelve subjects from two groups showed substantial consistency for partner preference over seven days in term 1 (Table 2; Fig 2). The number of subjects with a substantial consistency value increased to six in term 2. These results indicate that proximity to a partner was constant in some individuals and this tendency continued in term 2. In term 2, nine out of twelve subjects showed moderate to substantial consistency. Proximity to a partner was not necessarily to the previous cage mates in term 1. Differences in time spent in proximity to each member was not based on familiarity (N = 12, 18 dyad, LMM and likelihood ratio test on the effect of familiarity, term1: $\chi^2$ (1) = 2.188, p = 0.139, term2: $\chi^2$ (1) = 0.333, p = 0.564; S2 Fig). These findings indicate that partner preference was formed over time in female rats, and whether rats preferred familiar or unfamiliar partners depended on the individual.

## Correlation between measured personality and social interaction

Table 3 shows the correlation matrix between individual scores of four principal components and the indices of social interaction. The components found to be relevant to social interaction differed between terms 1 and 2. PC 1 and PC 2 showed a significant correlation with indices only in term 1, whereas PC3 showed a remarkable effect in term 2. PC4 showed a similar tendency in both terms 1 and 2.

In term 1, PC1, which was interpreted as exploration, was positively correlated with the time spent in proximity to unfamiliar group members. PC2, interpreted as boldness, also showed a weak positive correlation with that index. PC2 was negatively correlated with time spent in isolation. PC4, interpreted as tameness, was positively correlated with time spent in isolation, and negatively correlated with time spent in proximity to familiar members. PC3, interpreted as activity, was also negatively correlated with time spent in proximity to familiar members. Social relationships were less fixed in term 1 compared to term 2. In this phase, proximity with a prior cage mate and a newly introduced member were associated with different behavioral characteristics (PC1 and PC2 for unfamiliar individuals and PC3 and PC4 for familiar individuals, respectively). There was no principal component associated with indices for approach and avoidance in term 1.

PC3, interpreted as activity, was correlated with both proximity and avoidance behavior with familiar members in term 2. High-scoring individuals spent less time in proximity to a prior cage mate and were more likely to avoid or break proximity with a prior cage mate. PC4 also showed a negative correlation with proximity with familiar members. In term 2, where the social relationships were more stable compared to term1, behavioral characteristics were associated with a preference for familiar members, but not for unfamiliar members. Indices for approach were not strongly correlated with any components in terms 1 or 2.

## Similarity and preference

We examined the association between similarity to a partner's measured personality traits and preference for that partner. We found no strong correlation between the similarity of PC scores and time spent in proximity (Table 4). Therefore, at least the similarity of these four personality traits among partners was not associated with partner preference in our subject female rats.

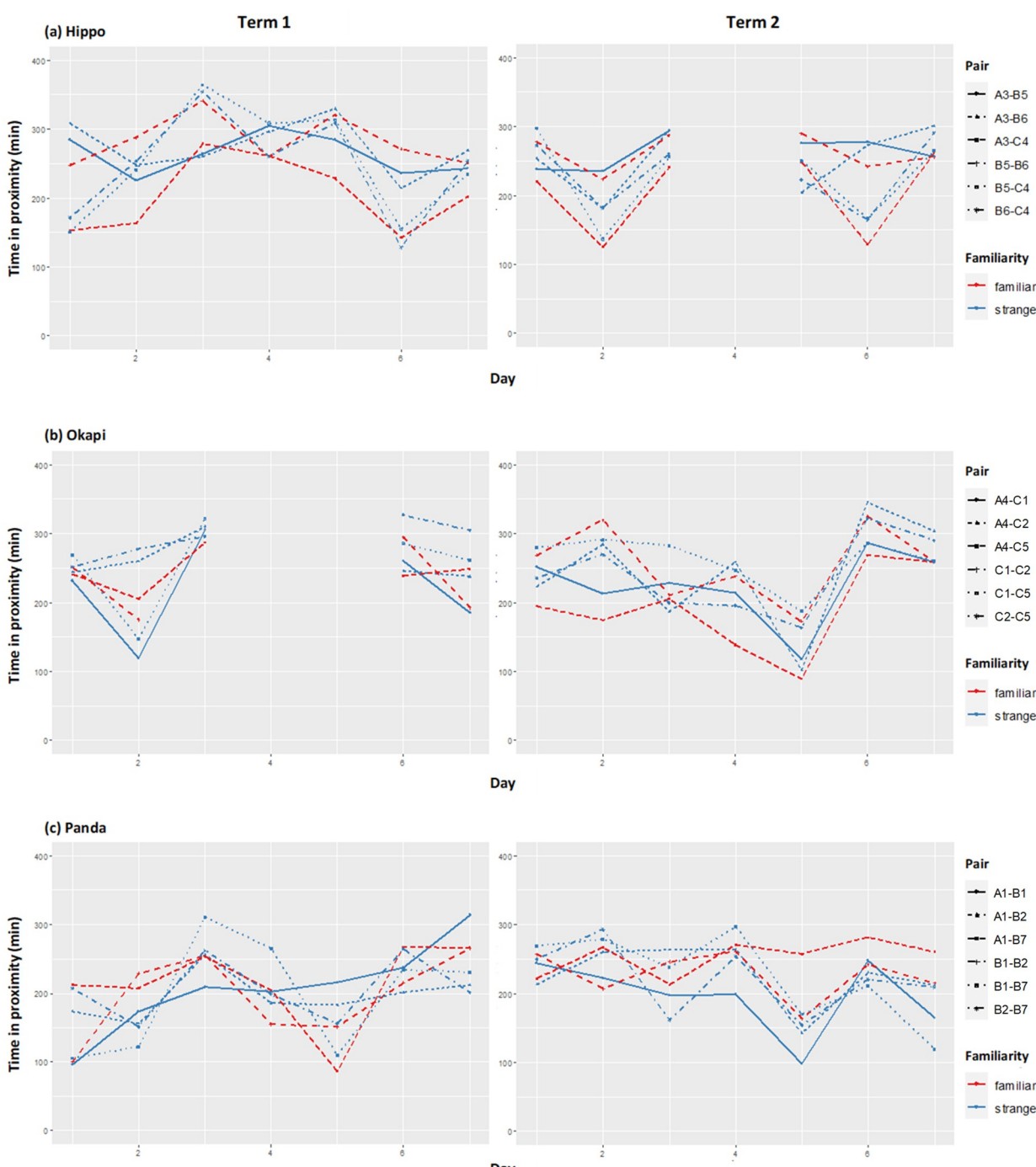

**Fig 2. Daily changes in time spent in proximity for each combination in three groups.** Red lines indicate the familiar pairs, and blue lines indicate the unfamiliar pairs. The following days were excluded from the analysis: Days 4 and 5 in term 1 for the Okapi group, and Day 4 in term 2 for the Hippo group.

**Table 3. Pearson correlation coefficients (r) between principal component values and the indices of social interaction standardized by group average value.**

| Social interactions | PC scores | | | |
|---|---|---|---|---|
| **Term 1** | **PC1** | **PC2** | *PC3* | **PC4** |
| Time in isolation | 0.070 | **-0.441** | 0.070 | **0.608** |
| Time in proximity: familiar | -0.141 | 0.057 | **-0.424** | **-0.466** |
| Time in proximity: unfamiliar | **0.594** | 0.336 | 0.161 | -0.308 |
| Approaching: familiar | -0.182 | -0.098 | 0.105 | 0.161 |
| Approaching: unfamiliar | -0.196 | -0.294 | -0.063 | 0.259 |
| Avoidance: familiar | -0.168 | 0.084 | 0.238 | 0.063 |
| Avoidance: unfamiliar | -0.133 | -0.371 | -0.084 | 0.280 |
| Term 2 | | | | |
| Time in isolation | 0.112 | 0.049 | **0.622** | 0.308 |
| Time in proximity: familiar | -0.339 | 0.297 | **-0.495** | **-0.410** |
| Time in proximity: unfamiliar | 0.189 | -0.133 | -0.084 | -0.364 |
| Approaching: familiar | -0.217 | 0.070 | 0.364 | -0.035 |
| Approaching: unfamiliar | -0.210 | -0.252 | 0.245 | -0.042 |
| Avoidance: familiar | 0.063 | 0.000 | **0.511** | 0.077 |
| Avoidance: unfamiliar | -0.133 | -0.161 | 0.350 | -0.056 |

Inverted positive/negative values were shown for PC3 (italicized) for ease of interpretation. Correlation coefficients with a value of ± 0.4 to 1.0 were interpreted as moderate to high correlation, and highlighted in boldface. The confidence intervals for the correlation coefficients are summarized in S3 Table.

## Discussion

### Personality structure of rats

Although laboratory rats are standardized domesticated animals, they do show inter-individual differences in behavior. The principal component analysis (PCA) based on the 13 variables from behavioral tests yielded four principal components. While it is important to note that there might be some overlaps between different behavioral tests in the traits they measure [51], in general, personality traits corresponding to each test were detected as principal components. This suggests that each behavioral test was able to measure the trait that it intended to measure.

Despite the rather ambiguous distinction between exploration-avoidance and boldness, we interpreted PC1 as the exploration-avoidance axis and PC2 as the shyness-boldness axis. The exploration-avoidance axis describes the individual's reaction to a novel situation, while boldness corresponds to the animal's reaction to a risky situation [17]. In our study, PC1 corresponds to the novel object test and PC2 was characterized by the elevated plus maze test. Although both tests are novel situations for rats and thus may relate to exploration, these

**Table 4. The Pearson correlation coefficients (r) between the similarity score of principal component values and time spent in proximity with that partner.**

| Time in proximity | Differences in absolute PC scores | | | |
|---|---|---|---|---|
| | **PC1** | **PC2** | *PC3* | **PC4** |
| Term1 | -0.226 | 0.121 | -0.115 | -0.094 |
| Term2 | -0.129 | 0.185 | -0.205 | -0.238 |

Inverted positive/negative values were shown for PC3 (italicized) for ease of interpretation. The values standardized by group average were used for time spent in proximity.

interpretations were adopted because the elevated plus maze has more anxiety-evoking characteristics, such as the perception of danger. Food that are generally highly palatable was used for the novel object test; however, it is unlikely that approach behavior in this test can simply be explained by hunger. Animals were not food deprived and did not consume the novel food. Some subjects even showed freezing behavior. Therefore, we consider that this test was able to measure individual differences in novelty exploration/avoidance.

We interpreted PC3 as activity level. PC3 was characterized by total distance traveled and center area entry in the open field test. It is noteworthy that the open field test is also used for measuring traits other than activity in rodents, such as exploration-avoidance [57, 58]. In our results, PC3 was detected as a separate component other than PC1 (exploration-avoidance) or PC2 (shyness-boldness). Also, distance traveled in the open field test, the item with the highest loading for PC3 is typically interpreted as activity [46]. Thus, the interpretation of PC3 as activity level seems reasonable.

PC4 could be interpreted as insensitivity. PC4 was characterized by time spent being handled in the glove test. Based on handling test studies in domesticated animals, those that tolerate handling longer are interpreted as having a high level of passive tameness [50, 59]. Also, subjects with high PC4 scores would often enter the center area in the open field test. Considering that both gloved human hands and open fields are potentially hazardous for rats, animals with high PC4 scores might be insensitive to threats or aversive situations.

Although we aimed at measuring sociability in the three-chamber test, sociability did not emerge as a significant principal component in PCA. Given that the three-chamber test is considered a well-established sociability test, this may be attributed to the specific experimental procedure employed in this study. For example, variable transformations were performed before PCA to increase the validity of the analysis, which may have made sociability less detectable. Also, it is possible that sociability is less likely to appear as a measured personality trait in laboratory rats, since they typically possess remarkable sociability and there could be less individual differences in this trait compared to others (i.e., exploration or boldness).

The personality structure detected in our study includes the exploration-avoidance axis, the shyness-boldness axis, activity level, and tameness. Studies in other species including rodents also report similar personality structures; that is, behavioral characteristics related to exploration, boldness, and activity level are detected as major personality components (bank voles, *Myodes glareolus* [60]; common marmosets, *Callithrix jacchus* [61]). In addition, although this study used only female rats, the results were comparable to those of the male rat study. For example, Rudolfová, et al. [46] report that the open field test and the elevated plus maze test represent separate axes, which is consistent with our data. Whether there are sex differences in the personality structure of rats is an issue that needs further investigation.

As we did not perform personality tests multiple times, we cannot confirm the repeatability of behavioral traits. Previous studies report that activity-related behavior (distance traveled) in rats show higher consistency than others [46, 62], so we assume that PC3 (activity level) in our results may be a consistent component.

## Individual differences in social relationships

We found consistency in the time spent in isolation, the index for sociability, but the degree of consistency differed based on the group. Sociality is reported as a robust trait in rats [62]. Although sociability measured by the three-chamber test did not appear in the four principal components, this finding suggests the presence of consistent sociability tendencies in our subject female rats.

Consistent partner preference was found in three out of twelve subjects in term 1, and five out of twelve subjects in term 2. In our study, group variations in personality traits may have affected the group differences in consistency. For example, the scores of PC3 were relatively high in the Panda group, compared to the other two groups.

Familiarity, the previous experience of cohabitation, did not explain this preference. Moreover, three familiar pairs were littermates and had been housed together from birth (B1-B2, B5-B6, and C1-C2), but were not necessarily the most preferred partner (Fig 2). This contrasts with many female mammals. Preference for kin has been reported in female mammals, who are mostly the philopatric sex [63]. A study on wild Norway rats living in highly urbanized areas did not show clear philopatry or sex-biased dispersal patterns [64]. Social bonds based on kin relations has been reported between mother-infant pairs but not between adults (see [65]). Moreover, previous studies on social bonding in captive rats reported inconsistent results; preference/avoidance association patterns were found in most individuals in males [44], but association patterns did not differ from chance, and were not consistent over time in females [45]. Combined with our results, social relationships in Norway rats and their domesticated species may have a more flexible, opportunistic nature.

In contrast to previous studies focused on the social relationships of established groups, our study tracked behavior from the first encounter of an unfamiliar member to the formation of the social relationship. We conducted the second phase of behavioral tracking three weeks after the new group was introduced. In experiments studying prosocial behavior or the social buffering effect, after about one to three weeks of living together, rats show differential behavior towards cage mates compared to unfamiliar individuals [39, 66, 67]. In term 2, approximately half of the subjects formed a preference for a partner. Overall, our study was able to observe social relationships after several partner preferences had formed. The differences in group size, strain, and ages of the subjects may explain the discrepancy seen with previous studies in female rats. In addition, different ways of evaluating social relationships could also have a significant impact on this; we focused on consistency within a short period. Whether long-term consistency in partner preference exists is an issue to be considered in the future. Nevertheless, our study was able to clarify whether female rats prefer certain cage mates under some conditions, by focusing on the formation phase of social relationships.

## Relationships between measured personality traits and social interactions in a group

We exploratory examined the relationship between measured personality traits and social interactions and found that the different components were associated with the social interactions in two phases: formation and maintenance of social relationships.

In term 1, the exploration-avoidance and shyness-boldness axes were associated with indices related to proximity. Thus, these traits are related to individual differences in partner preference or preference for being alone during the relationship formation phase. Since these characteristics were evaluated based on reaction to risky or novel situations, it is quite possible that these were more likely to influence behavior in an environment with unfamiliar individuals, although the subjects were habituated to the housing environment prior to social tracking.

In term 2, activity level was conspicuous and nearly the only components that showed high correlations. The subjects were accustomed to group members in this phase, so initially unfamiliar members were no longer novel. It is possible that the influence of the exploration-avoidance and shyness-boldness axes was progressively smaller, and the influence of activity level was greater, instead. A study on laboratory rats reported activity was consistent across time

and context [62]. Given this, it seems reasonable that the influence of activity strongly appeared after acclimation to the physical and social environment.

Our results were not always consistent with previous studies on personality traits and the patterns of social interaction. It has been reported in several species that the exploration is positively correlated with sociability [26], and bold individuals are less sociable (e.g., [21]). This discrepancy might be because we also conducted observation during the formation phase in our study, unlike previous studies. In support of this, results on activity level in term 2 were consistent with previous studies in several species that less active individuals spend more time with others [68]. This is thought to arise because rats tend to share their home-base with other rats, and the least active rat, which tends to be stationary, is gradually joined by other group members [69]. Passive tameness, evaluated by low aversiveness to humans, was related to time spent in isolation and proximity to familiar members. To our knowledge, this is the first report examining the relationship between individual differences in tameness and social behavior among rats. Passive tameness is a characteristic considered in the context of domestication. When animals, including rats, are selectively bred for tameness, the domesticated animals show reduced stress hormone levels [70, 71]. Given this, individuals with higher tameness should be likely to be less stressed when separated from other individuals, although it should be noted that the previous studies examined differences in strain, not individual differences in tameness. Finally, there may be sex differences in association between the personality traits and social interactions, as the previous study showed a correlation between boldness, activity and sociability in male guppies, but not in females [24]. Considering the sex differences in social relationships in rats, especially pronounced dominance hierarchy in males [65], experiments on a male population might produce different results than the current one.

Similarity in personality traits between two group members was not associated with proximity to that member. In a previous study, similarity of sociability was mainly related to proximity with that individual (e.g., [32]), but we could not detect sociability in the principal components analysis. The personality measures may need to be improved to clarify whether homophily is present in rats.

It is reasonable to assume that personality traits are associated with social behavior patterns, because these personality traits can be viewed as individual differences in response to acute or chronic stress caused by the social or physical environment, such as an encounter with an unfamiliar individual, or changes in group composition, habitat, or access to limited resources. For example, a familiar partner more effectively plays a role for stress buffering [66]. Individuals with lower stress resistance may spend more time in proximity to familiar individuals, when under such acute stress. We propose the following hypothesis based our results: In the formation phase, personality traits related to changes in the physical and social environment influence proximity relationships, because boldness and exploration reflects the reaction to a novel, or potential risky situation [17]. Then, in the maintenance phase, activity level, which is a measure in more general situation, predicts the pattern of social behaviors. Balanced group formation based on personality traits, and a group size large enough to adequately measure individual preferences will allow for appropriate testing of these hypotheses.

The total number of approaches/avoidances showed little or no association with personality traits. In future studies, social play and aggressive interactions should be used as behavioral indicators, rather than simply approaches or avoidances.

## Future directions

Since the population size in this experiment was four individuals per group, there were not many options for relationship formation. If we could conduct the experiment with a larger

group size, we may be able to see a clearer correlation between individual characteristics and inter-individual relationships.

Even though we observed social relationship building over a very short period, the correlations between inter-individual relationships and individual characteristics in term 2 changed from those in term1: the impact of components with a relatively large proportion of variance (PC1 and PC2) decreased and the influence of PC3 increased instead. It is suggested that the accumulation of social interactions may change individual characteristics or bias the weight of variables explaining individual behavior toward social relationships rather than individual characteristics. Future studies should analyze the dynamics between individuals and the social environment by measuring behavioral traits over longer periods of time and observing group behavior.

In this experiment, we did not observe a clear social ranking such that dominance could be measured. This may be because female rats were used as experimental subjects. Although some results have shown no correlation between social rank and social interaction in rats [44, 45], it may be possible to analyze the correlation between inter-individual relationships and social rank in male subjects.

Even though rats are animals that have been selectively bred as model animals, the present experiment showed a variety of behavioral characteristics. This could benefit experimental psychology, if such variations influence how rats complete experimental tasks, particularly willingness to complete tasks, and the rate of completion. If it becomes possible to ensure consistency in behavioral characteristics when selecting individuals for experiments, it will be possible to conduct more controlled behavioral experiments.

It is difficult to observe long-term social interactions among rats using DLC individual tracking techniques. This is because rats, which are nocturnal, are difficult to track when photographed in the dark. In addition, tracking sites can be lost due to their flexible movements or when they jostle one another. In this study, we overcame this problem by using a tracking point correction program, enabling long-term tracking of individuals in the dark. It is now possible to study the development and environmental adaptability of rats as model animals. In the future, we aim to conduct more detailed analysis by also implementing posture estimation and behavior coding.

## Supporting information

**S1 Fig. Scree plot derived from the PCA for the items of five behavioral tests.**
(TIF)

**S2 Fig. Differences in preference based on the familiarity with the group members.** There were no significant effects of familiarity (N = 12, 18 dyad, LMM and likelihood ratio test, term1: $\chi2$ (1) = 2.188, p = 0.139, term2: $\chi2$ (1) = 0.333, p = 0.564).
(TIF)

**S1 Table. The result of the position plot test.**
(TIF)

**S2 Table. Group differences in total time spent in proximity to other members in term 1 and term 2.** Significant differences by LMMs and likelihood ratio test was found in term 1 (N = 12).
(TIF)

**S3 Table. The 95% confidence intervals of correlation coefficients between principal component values and the indices of social interactions standardized by group average value.**

Inverted positive/negative values were shown for PC3 for ease of interpretation.
(TIF)

**S1 File.**
(DOCX)

## Acknowledgments

We thank members of the Okanoya laboratory at the University of Tokyo and Teikyo University for stimulating discussion. We are grateful to Dr. Beth A. Vernaleo for her careful proof reading of this paper. Tanamove, the application used to analyze the glove test, was developed by Akira Tanave.

## Author Contributions

**Conceptualization:** Shiomi Hakataya, Noriko Katsu, Genta Toya.

**Data curation:** Shiomi Hakataya, Noriko Katsu, Genta Toya.

**Formal analysis:** Shiomi Hakataya, Noriko Katsu, Genta Toya.

**Funding acquisition:** Shiomi Hakataya, Noriko Katsu, Kazuo Okanoya, Genta Toya.

**Investigation:** Shiomi Hakataya, Noriko Katsu, Genta Toya.

**Methodology:** Shiomi Hakataya, Noriko Katsu, Genta Toya.

**Project administration:** Shiomi Hakataya, Noriko Katsu, Genta Toya.

**Resources:** Shiomi Hakataya, Noriko Katsu, Kazuo Okanoya, Genta Toya.

**Software:** Genta Toya.

**Supervision:** Kazuo Okanoya.

**Validation:** Shiomi Hakataya, Noriko Katsu, Kazuo Okanoya, Genta Toya.

**Visualization:** Shiomi Hakataya, Noriko Katsu, Genta Toya.

**Writing – original draft:** Shiomi Hakataya, Noriko Katsu, Genta Toya.

**Writing – review & editing:** Kazuo Okanoya.

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
