## [Decision Letter · Decision Letter 0]

25 Sep 2023

PONE-D-23-27375An exploratory study of behavioral traits and the establishment of social relationships in female laboratory ratsPLOS ONE

Dear Dr. Hakataya,

Thank you for submitting your manuscript to PLOS ONE. After careful consideration, we feel that it has merit but does not fully meet PLOS ONE’s publication criteria as it currently stands. Therefore, we invite you to submit a revised version of the manuscript that addresses the points raised during the review process.

We look forward to receiving your revised manuscript.

Kind regards,

Alexandra Kavushansky, PhD

Academic Editor

PLOS ONE

Journal Requirements:

2. Please expand the acronym “JSPS” (as indicated in your financial disclosure) so that it states the name of your funders in full.

Additional Editor Comments:

This is an interesting and important study. I have several questions:

1. The authors state that due to differences in the times of recording sessions in the behavioral tests, the indexes related to time (e.g. time spent near the novel object) were not included in the analysis. However, this raises a question of the interpretability of the results. For instance, if the animal spends a lot of its time near the novel object exploring it, its interest in the object will not be reflected in either distance travelled in the maze, or in number of entries into the object’s zone. Similarly, indexes of distance travelled and number of entries in other tests, such as in social behavior in the 3-chamber apparatus, may not be reflective of the actual behavior measured. The technical issue of differences between the total times of recording the sessions may be overcome by calculating the time spent in each behavior as a proportion of the total time of the session. Please validate your findings by examining the proportional times of each behavior.

2. Similarly, a question of interpretability arises by the choices in the way of performing the tests. Specifically, why in the Novel object test (NOT), the object itself was food? Especially, considering the fact that the animals were food deprived before the testing? In this test aimed to check neophobia/neophilia, usually a neutral object is used. In the current study setup, a hungry animal smelling a chocolate-coated puff, would naturally be driven to it by hunger itself, not curiosity. Even a non-deprived rat smelling chocolate would rush to check it. Please explain the rationale behind such a choice of the novel object.

3. Line 166 – please change the “names” to “named”.

Reviewers' comments:

Reviewer's Responses to Questions

**Comments to the Author**

1. Is the manuscript technically sound, and do the data support the conclusions?

Reviewer #1: Yes

Reviewer #2: Yes

2. Has the statistical analysis been performed appropriately and rigorously? 

Reviewer #1: Yes

Reviewer #2: Yes

3. Have the authors made all data underlying the findings in their manuscript fully available?

Reviewer #1: Yes

Reviewer #2: Yes

4. Is the manuscript presented in an intelligible fashion and written in standard English?

Reviewer #1: Yes

Reviewer #2: Yes

5. Review Comments to the Author

Reviewer #1: Recommendation:

Minor revisions

MS#: PONE-D-23-27375

Title: An exploratory study of behavioral traits and the establishment of social relationships in female laboratory rat

Comments to Author:

I found this study on how personality traits of individual rats affect the establishment of social relationships very interesting. However, I did find the methodology a bit complex and it was easy to get lost in it. With a minor revision, mainly focusing on shortening and clarifying the introduction and methods, I believe the manuscript can be well-received by the broad audience of PLOS ONE.

Overview and general recommendation:

Major comments:

1. The introduction seems to have too much information that could potentially be shortened. One way to do this would be to combine and summarize the information under the subheading into a brief paragraph.

2. It seems like the methods section could be shortened, as there is some information repeated under different sections. Additionally, the section can be slightly confusing for the reader in terms of understanding which phase of housing each test is occurring in. One way to make this clearer would be to indicate in Figure 1 which phase of housing each test is occurring in. For example, the glove test occurs in pair housing 2, and so on. This would help the reader better understand the results.

Minor comments:

Abstract

Line 25: Please add what species of rat and the species name.

Introduction

Comment 1: Remove subheadings.

Line 101: What is a fancy rat?

Line 123; Add species name for guinea pigs.

Methods

Comment 1: This section is slightly confusing please indicate of Figure 1 in which phase of housing is each test is occurring. For example glove test occure in pair housing 2, ect.

Line 159: What happened to the remaining 2 animals?

Line 324: What is the word term referring to?

Discussion:

Line 611: Give examples of how the physical and social environment influence proximity

relationships. Do the same for Line 613.

Figures:

Figure resolution is poor in the version I received. This has made it hard to comment on them.

Final remarks

I hope these comments are useful to the authors. I would be very happy to peruse the manuscript again after revision. I congratulate the authors on a very interesting paper.

Reviewer #2: I’d like first to thank the authors for the opportunity to read their work. The paper is clear and informative and the study makes an important contribution to the field. The methods were appropriate and the discussion was compatible with the results. However, some relevant topics and concepts should be revised and expanded especially in the introduction. On the attatched file, I list my suggestions/questions point-by-point. I hope these remarks will help the authors improve their manuscript.

6. PLOS authors have the option to publish the peer review history of their article (what does this mean?). If published, this will include your full peer review and any attached files.

Reviewer #1: No

Reviewer #2: No

---

## [Author Response · Author response to Decision Letter 0]

19 Oct 2023

Replies to comments from the editor

We sincerely appreciate the constructive comments, which helped to improve our manuscript substantially. We addressed all comments made by the editor and reviewers. We also went over the entire manuscript again and made some corrections where necessary. Below, we respond point-by-point to the comments raised by the editor.

Comments from the editor

2. Please expand the acronym “JSPS” (as indicated in your financial disclosure) so that it states the name of your funders in full.

Reply:

- The three comments above are addressed within the cover letter to the editor.

Additional Editor Comments: This is an interesting and important study. I have several questions:

1. The authors state that due to differences in the times of recording sessions in the behavioral tests, the indexes related to time (e.g. time spent near the novel object) were not included in the analysis. However, this raises a question of the interpretability of the results. For instance, if the animal spends a lot of its time near the novel object exploring it, its interest in the object will not be reflected in either distance travelled in the maze, or in number of entries into the object’s zone. Similarly, indexes of distance travelled and number of entries in other tests, such as in social behavior in the 3-chamber apparatus, may not be reflective of the actual behavior measured. The technical issue of differences between the total times of recording the sessions may be overcome by calculating the time spent in each behavior as a proportion of the total time of the session. Please validate your findings by examining the proportional times of each behavior.

Reply:

- This is a very important concern, however, the time measure was not reliable because the number of frames per second varied within the same video by accident (Line 232). Therefore, we believe it is inappropriate to use time-related indicators in PCA. For reference, there were high positive correlations between duration in a specific are and entry times to that (Table below), though it should be noted measurements related to duration are imprecise.

2. Similarly, a question of interpretability arises by the choices in the way of performing the tests. Specifically, why in the Novel object test (NOT), the object itself was food? Especially, considering the fact that the animals were food deprived before the testing? In this test aimed to check neophobia/neophilia, usually a neutral object is used. In the current study setup, a hungry animal smelling a chocolate-coated puff, would naturally be driven to it by hunger itself, not curiosity. Even a non-deprived rat smelling chocolate would rush to check it. Please explain the rationale behind such a choice of the novel object.

Reply:

- We realize that this explanation was not sufficient in the previous version. We have added the reason for using chocolate puffs (Line 267-268). When we used neutral objects in a preliminary experiment on rats other than the study subjects, they hardly showed approaching behaviors. Therefore, we used food potentially high value for rats.

- We also added discussion of whether this test could measure novelty exploration/avoidance (Line 497-500). 

3. Line 166 – please change the “names” to “named”.

Reply:

- Corrected, thank you (Line 198). 

Replies to comments from Reviewer #1

Comments to Author:

I found this study on how personality traits of individual rats affect the establishment of social relationships very interesting. However, I did find the methodology a bit complex and it was easy to get lost in it. With a minor revision, mainly focusing on shortening and clarifying the introduction and methods, I believe the manuscript can be well-received by the broad audience of PLOS ONE.

Reply:

- We sincerely appreciate the constructive comments, which helped to improve our manuscript substantially. We addressed all comments made by the editor and reviewers. Introduction and Methods have been rewritten throughout. We also went over the entire manuscript again and made some corrections where necessary. Below, we respond point-by-point to the comments raised by Reviewer #1. 

Major comments:

1. The introduction seems to have too much information that could potentially be shortened. One way to do this would be to combine and summarize the information under the subheading into a brief paragraph.

Reply:

- Thank you for bringing up this issue. We have deleted subheading and rewritten the introduction. However, we were eventually not able to shorten the introduction section because we needed to respond to reviewer 2's suggestions (clarifying limitations of personality axes approach and explaining each personality trait in detail). Instead, we made the method section shorter and easier to understand.

2. It seems like the methods section could be shortened, as there is some information repeated under different sections. Additionally, the section can be slightly confusing for the reader in terms of understanding which phase of housing each test is occurring in. One way to make this clearer would be to indicate in Figure 1 which phase of housing each test is occurring in. For example, the glove test occurs in pair housing 2, and so on. This would help the reader better understand the results.

Reply:

- Information repeatedly written under different parts were combined and deleted. Figure 1 was modified to make the timeline of experiments clearer. Also, we descried the flow of the study (what experiment happened when, housing conditions, and animals’ ages) in more detail in the experimental design section (Line 181-205).

Minor comments:

Abstract

Line 25: Please add what species of rat and the species name.

Introduction

Comment 1: Remove subheadings.

Line 101: What is a fancy rat?

Line 123; Add species name for guinea pigs.

Reply:

- These points have been amended (Line25; Line 114-115; Line 136-137). As mentioned above, subheadings were removed.

Methods

Comment 1: This section is slightly confusing please indicate of Figure 1 in which phase of housing is each test is occurring. For example glove test occure in pair housing 2, ect.

Reply:

- As above, Figure 1 was modified to make the timeline of experiments clearer.

Line 159: What happened to the remaining 2 animals?

Reply:

- We have added some more information in Line 188-191.

Line 324: What is the word term referring to?

Reply:

- Thank you for pointing out. The previous manuscripts used the ‘term’ without explanation. We have added the information in the experimental design section (Line 195-196, 201-203) and in Figure 1.

Discussion:

Line 611: Give examples of how the physical and social environment influence proximity

relationships. Do the same for Line 613.

Reply:

- We have added examples on how differences in physical and social environment can affect social relationships (Line 611-621).

Figures:

Figure resolution is poor in the version I received. This has made it hard to comment on them.

Reply:

- We are not sure why, but it seems that when the system converted the file to PDF, the resolution of the file was reduced. We would appreciate if you could check the original files. The original files can be downloaded from the link in the upper right corner of the figure page in manuscript PDF.

Final remarks

I hope these comments are useful to the authors. I would be very happy to peruse the manuscript again after revision. I congratulate the authors on a very interesting paper. 

 

Replies to comments from Reviewer #2

Reviewer #2: I’d like first to thank the authors for the opportunity to read their work. The paper is clear and informative and the study makes an important contribution to the field. The methods were appropriate and the discussion was compatible with the results. However, some relevant topics and concepts should be revised and expanded especially in the introduction. On the attatched file, I list my suggestions/questions point-by-point. I hope these remarks will help the authors improve their manuscript.

Reply:

- We sincerely appreciate the constructive comments, which helped to improve our manuscript substantially. We addressed all comments made by the editor and reviewers. Introduction has been rewritten throughout. The reason and the possible effect of choice for using solely female rats have been added. We also went over the entire manuscript again and made some corrections where necessary. Below, we respond point-by-point to the comments raised by Reviewer #2. 

Page 4, line 48: The authors should specify which kind of social bonds are reported here. For example, parenting will have a different effect on the longevity of female baboons than bonds based on conflicts. 

Reply:

- We are not sure we understand exactly what the reviewer intended, but we specified as “Strong affiliative social relationships between same-sex individuals” (Line 46). 

Page 5, line 59: The approach of the authors to personality is rather too simple and distant from current theories. Personality is now understood as more than just a set of timely-conserved behaviors (see Lages and McNaughton, 2022; Revelle, 2007) as bottom-up and top-down approaches are often combined in attempts to characterize personality in non-human animals and in translational research (for a background: Big-5, HiTOP, and Eysenck/Gray/RST theories). The authors should care for a more detailed approach of Personality and then define the scope in which they will work with. 

Page 5, line 59: While these axes are certainly used to characterize personality traits in non-human animals, they are not the only ones. For example, while in fishes these axes can characterize well populations, they may not be suitable in describing individuals (see Carter et al., 2013; Sánchez-Tójar et al., 2022). On the other hand, these axes may be too simplistic to describe the personality of rodents (see Michelini, G, 2021). The authors should cite other theories of organization and then concentrate on their chosen approach while acknowledging its limitations. 

Reply:

- Thank you for pointing out. While we recognize that various other approaches exist to characterize personality, we chose the simple 5-axis paradigm (Réale, et al., 2007) because it enables comparison with many other non-primate animal studies. We have added some explanations about other personality study approaches (Line 57-65) and limitations of our approach (Line 68-72).

Page 5, line 63: Although a very brief explanation of each axis is given in parenthesis, in the following subsections they must be defined and characterized more thoroughly. 

Page 5, line 71: As stated before, the text can benefit from a more descriptive explanation of the meaning of each term (in this case, exploration and activity). 

Reply:

- We have added explanation of each axis in the following sections: Boldness (Line 73-74); Exploration and activity (Line 80-82); Sociability (Line 89-90).

Page 6, page 82: it lacks the species nomenclature of “Eastern garter snakes”. 

Reply:

- “Eastern garter snakes” appeared previously (Line 76), so scientific names were not indicated this time (Line 95). We specified “Eastern” on the first time it appeared.

Page 6, line 82: As the authors did in the topics before, here the text could benefit from citation of results in different species. Also, citations of papers comparing natural observations x experimental approaches, as this was a topic approached earlier by the authors. 

Reply:

- We apologize for having not been clear, but the description in line 82 (previous manuscript) is studies on two different species, using the behavioral assay and observation under the natural setting. We specified that in Line 93-96.

Page 8, line 125: Do differences in tameness in lab animals reflect differences between wild and (general populations of) lab animals? If the authors chose to use this measure, this correlation should be showed before. 

Reply:

- Thank you for bringing up the issue. We have added arguments that individual differences in tameness can be measured within the same strain (Line 138-140).

Page 9, line 133: While the choosing of these tests is adequate in my point of view, the authors should acknowledge the dependence between the difference measures. For example, the open field test was used for measure activity, however these results can also be impacted by boldness. None of the tests is "clean" enough to be a reflection of just one personality trait. It can be stated in the introduction and then discussed taking into account the results obtained with the principal components analysis (as the authors have already started doing). 

Reply:

- Thank you for your important suggestions. We mentioned the dependence between the behavioral tests and related concerns in the introduction (Line 149-153) and discussion (Line486-488, 494-496, 502-503). 

Page 10, line 151: Why only females were used? The authors should justify it in the introduction (or alternatively in methods). 

Reply:

- We realize that explanation was insufficient in the previous manuscript. We have added information in Methods (Line 178-179). We used solely female rats because we aimed to examine social interactions between same-sex individuals outside breeding context. 

Page 10, line 166: Typo – “names” > “named”. 

Reply:

- Corrected, thank you (Line 198).

Page 13, line 203: Please specify how many hours of food deprivation. 

Reply:

- Food deprivation was approximately four hours (Line 227-228). We removed food from the home cage at 9:00 am, so the precise duration of the deprivation depended on the test order of the subject. The duration of deprivation between dyad was same. 

Page 14, line 224: “first” can be changed to “at first”. 

Page 14, line 225: “second” can be changed to “later” or “next”. 

Reply:

- These points were changed as suggested (Line 261 and 262).

Page 20, line 336: To which tests were the DLC analysis applied to? All? The authors should specify.

Reply:

- We stated it in the behavioral tracking section (Line 335-338). 

Page 30, line 466: Although not reaching the cutoff criteria, comments can be made about the association of PC2 with proximity to unfamiliar groups, as it is related to exploration and boldness itself (as the authors correctly pointed out in the discussion). 

Reply:

- Thank you for an important suggestion. We specified that point in Line 452-454 and 459.

Page 33, line 521: Which other inferences can one have with the three-chamber test? Do the authors believe this test should not be used to test sociability in general or is this a particularity of their experimental setup? The authors should make a comment about it, especially because this test has been classically used for measuring sociability (see Rein B, 2020; Kaidanovich-Beilin, 2011) 

Reply:

- As in results (Line 400-402), we interpreted sociability measured by this test did not explain the large variance of behavioral characteristics, compared to other characteristics. We have added some more explanation for this in the discussion (Line 513-520). 

Page 36, line 563: Not only here, but throughout the discussion the authors must comment on their choice for using solely female rats and how their results could be extrapolated (or not) to both sexes. 

Reply: 

- We consider that point important. The results that should be mentioned in reference to the use of females, has been amended (Line 409-411, 427-428, 473-474). We have also added some discussion on the possible effect of sex (Line 525-529, 601-606).

Page 37, line 583: As reasonable as this hypothesis seems, the text could be improved with citations that show the decrease of the influence of the exploration-avoidance and shyness-boldness reduced over time.

Reply:

- As far as we know, there was no references on the effects on social interactions decrease over time. Instead, we have added explanation that exploration-avoidance and shyness-boldness reflect individual reaction to novel and/or risky situations, and relationship formation phase is such situations. Corresponding modifications are in Line 611-621.

---

## [Decision Letter · Decision Letter 1]

20 Nov 2023

An exploratory study of behavioral traits and the establishment of social relationships in female laboratory rats

PONE-D-23-27375R1

Dear Dr. Hakataya,

We’re pleased to inform you that your manuscript has been judged scientifically suitable for publication and will be formally accepted for publication once it meets all outstanding technical requirements.

Kind regards,

Alexandra Kavushansky, PhD

Academic Editor

PLOS ONE

Additional Editor Comments (optional):

Reviewers' comments:

Reviewer's Responses to Questions

**Comments to the Author**

1. If the authors have adequately addressed your comments raised in a previous round of review and you feel that this manuscript is now acceptable for publication, you may indicate that here to bypass the “Comments to the Author” section, enter your conflict of interest statement in the “Confidential to Editor” section, and submit your "Accept" recommendation.

Reviewer #1: All comments have been addressed

Reviewer #2: All comments have been addressed

2. Is the manuscript technically sound, and do the data support the conclusions?

Reviewer #1: Yes

Reviewer #2: Yes

3. Has the statistical analysis been performed appropriately and rigorously? 

Reviewer #1: Yes

Reviewer #2: Yes

4. Have the authors made all data underlying the findings in their manuscript fully available?

Reviewer #1: Yes

Reviewer #2: Yes

5. Is the manuscript presented in an intelligible fashion and written in standard English?

Reviewer #1: Yes

Reviewer #2: Yes

6. Review Comments to the Author

Reviewer #1: (No Response)

Reviewer #2: I believe the manuscript has greatly improved since the first version. All my concerns have been properly addressed. I have no further comments at this point. I'd like to thank the authors for the opportunity to read their work.

7. PLOS authors have the option to publish the peer review history of their article (what does this mean?). If published, this will include your full peer review and any attached files.

Reviewer #1: No

Reviewer #2: No

---

## [Editor Report · Acceptance letter]

24 Nov 2023

PONE-D-23-27375R1 

An exploratory study of behavioral traits and the establishment of social relationships in female laboratory rats 

Dear Dr. Hakataya:

I'm pleased to inform you that your manuscript has been deemed suitable for publication in PLOS ONE. Congratulations! Your manuscript is now with our production department. 

Kind regards, 

on behalf of

Dr. Alexandra Kavushansky 

Academic Editor

PLOS ONE